# The First Highly Contiguous Genome Assembly of Pikeperch (*Sander lucioperca*), an Emerging Aquaculture Species in Europe

**DOI:** 10.3390/genes10090708

**Published:** 2019-09-13

**Authors:** Julien Alban Nguinkal, Ronald Marco Brunner, Marieke Verleih, Alexander Rebl, Lidia de los Ríos-Pérez, Nadine Schäfer, Frieder Hadlich, Marcus Stüeken, Dörte Wittenburg, Tom Goldammer

**Affiliations:** 1Institute of Genome Biology, Leibniz Institute for Farm Animal Biology (FBN), 18196 Dummerstorf, Germany; nguinkal@fbn-dummerstorf.de (J.A.N.); verleih@fbn-dummerstorf.de (M.V.); rebl@fbn-dummerstorf.de (A.R.); schaefer@fbn-dummerstorf.de (N.S.); hadlich@fbn-dummerstorf.de (F.H.); 2Institute of Genetics and Biometry, Leibniz Institute for Farm Animal Biology (FBN), 18196 Dummerstorf, Germany; perez@fbn-dummerstorf.de (L.d.l.R.-P.); wittenburg@fbn-dummerstorf.de (D.W.); 3State Research Center of Agriculture and Fisheries M-V, 17194 Hohen Wangelin, Germany; m.stueeken@lfa.mvnet.de

**Keywords:** genome assembly, genes annotation, pikeperch, fish, genome sequencing, aquaculture

## Abstract

The pikeperch (*Sander lucioperca*) is a fresh and brackish water Percid fish natively inhabiting the northern hemisphere. This species is emerging as a promising candidate for intensive aquaculture production in Europe. Specific traits like cannibalism, growth rate and meat quality require genomics based understanding, for an optimal husbandry and domestication process. Still, the aquaculture community is lacking an annotated genome sequence to facilitate genome-wide studies on pikeperch. Here, we report the first highly contiguous draft genome assembly of *Sander lucioperca*. In total, 413 and 66 giga base pairs of DNA sequencing raw data were generated with the Illumina platform and PacBio Sequel System, respectively. The PacBio data were assembled into a final assembly size of ~900 Mb covering 89% of the 1,014 Mb estimated genome size. The draft genome consisted of 1966 contigs ordered into 1,313 scaffolds. The contig and scaffold N50 lengths are 3.0 Mb and 4.9 Mb, respectively. The identified repetitive structures accounted for 39% of the genome. We utilized homologies to other ray-finned fishes, and ab initio gene prediction methods to predict 21,249 protein-coding genes in the *Sander lucioperca* genome, of which 88% were functionally annotated by either sequence homology or protein domains and signatures search. The assembled genome spans 97.6% and 96.3% of Vertebrate and Actinopterygii single-copy orthologs, respectively. The outstanding mapping rate (99.9%) of genomic PE-reads on the assembly suggests an accurate and nearly complete genome reconstruction. This draft genome sequence is the first genomic resource for this promising aquaculture species. It will provide an impetus for genomic-based breeding studies targeting phenotypic and performance traits of captive pikeperch.

## 1. Introduction

The Percidae family is a diverse and economically important group of mostly freshwater fishes that comprises 11 genera and about 275 identified species [1]. Many of these species play key roles in aquatic ecosystems and some provide valuable resources for aquafarming in recirculating aquaculture systems (RAS), which are a modern and ecologically viable alternative to ponds. Pikeperch (*Sander lucioperca* L., 1758, NCBI taxonomy ID: 283035) is one of the highly valued fish species for both recreational and commercial fishing in Europe [2]. Its faster growth compared to other Percids, and its resilience and diversification potential make *Sander lucioperca* an attractive species for intensive rearing, as these traits are crucial for the potential yields in commercial production. While the global capture production of pikeperch has halved since 2010, its aquaculture production has increased two fold in the same time and exceeded 900 tons a year (Food and Agriculture Organization (FAO), 2018). This illustrates the increasing consideration of pikeperch for commercial aquafarming, but also suggests that pikeperch is a niche-market species. The native range of *Sander lucioperca* includes the Caspian, Black, Aral and Baltic Sea drainages, where they inhabit brackish waters. Meanwhile this species has been anthropogenically introduced to most regions in Europe, Northern America and Asia [3,4], making it the Percid species with the largest geographic expanse [5].

The size of the pikeperch haploid genome was estimated to be 1.14 pg (i.e., 1114 Mb) utilizing cytometric methods [6]. A diploid number of 48 (2n = 48) chromosomes was reported for this species [7,8]. Previous studies have also reported a XY/XX sex chromosomes system in the Percidae fish family [9]. As an emerging Percid for rearing systems, pikeperch shows a relatively high susceptibility to stress under captive conditions [10,11,12], which implies a reduced immune system, and thus sensitivity to pathogens as a corollary. Furthermore, intra-cohort cannibalism in early life stages [13], is one of the major issues while rearing pikeperch. Population genetic studies on pikeperch could reveal molecular markers that are associated with juvenile cannibalism and predation avoidance. However, genomic data to conduct such genome-wide studies and genome-based selection for economical traits are currently lacking.

In the present study, we report the first highly contiguous and nearly complete draft assembly of the *Sander lucioperca* genome—constructed using long read sequencing by PacBio and taking advantage of accurate Illumina short reads to improve the base-level quality of the assembly and gene prediction reliability. We applied different approaches to evaluate the assembly including read alignment statistics, gene space statistics and comparative alignments with other teleosts. This draft assembly provides a valuable genomic tool to facilitate genome-wide research on pikeperch and the identification of functional markers associated with relevant commercial traits.

## 2. Materials and Methods

### 2.1. Sample Collection, Library Preparation

The sample tissues were obtained from a single adult male *Sander lucioperca*, collected in the state’s aquaculture facilities in Hohen Wangelin, Germany. Genomic DNA was extracted from liver, muscle and spleen tissues, which had been previously isolated and stored in liquid nitrogen. All DNA samples were pooled for the library’s preparation. For whole genome sequencing, we used multiple types of libraries. One short insert (paired-end, 470 bp) shotgun library was prepared using Illumina’s TruSeq DNA PCR-free library preparation kit. In addition, two size-selected mate-pair libraries with 2–8 kb and 2–10 kb long inserts were prepared following the Nextera mate pair library preparation protocol. To overcome the limitations of short reads for the assembly of complex eukaryote genomes, 20 kb large-insert PacBio libraries were also prepared according to the guide for preparing the SMRTbell template for sequencing on the PacBio Sequel System.

### 2.2. Whole Genome Sequencing, Quality Control

The size selected 20 kb DNA libraries were pooled and sequenced in 10 single-molecule real-time sequencing (SMRT) Cells on the PacBio Sequel II systems according to the SMRT^®^ sequencing guide. In total, 66 Gb of raw data accounting for 6.4 million polymerase reads were generated. Polymerase reads were trimmed using SMRT Link v6.0.0 to obtain 5.2 million high quality subreads (Appendix A). Additionally, one paired-end and two mate-pair libraries were sequenced on Illumina HiSeq X Ten platforms. The short insert size was on average 470 bp, while long inserts ranged from 2 to 10 kb.

To check the overall quality of sequencing data, FastQC vers.0.11.8 [14] was applied. Trimmomatic vers.0.38 was used to trim adapters, filter low quality reads (Q>28) and discard reads shorter than 40 bp. Since the mate-pair reads contained overrepresented sequences (0.1–1.2%), which probably originated from a TruSeq adapters contamination, we iteratively removed these overrepresented sequences using fastp vers.0.19.3 [15].

### 2.3. K-mer Based Genome Characteristics Estimation

Extensive knowledge of basic genome properties such as genome size, repeat content, and heterozygosity rate, supports the decision for an appropriate assembly strategy and the adequate parameters tuning. *K*-mer analysis is an efficient assembly-independent approach to estimate these genome characteristics prior to assembly. To estimate the *Sander lucioperca* genome size, we generated *k*-mer profiles from high-quality genomic paired-end reads using the program jellyfish vers.2.2.10 [16]. As applied in previous publications [17,18,19,20], the genome size *G* was calculated based on the following formulas: N=M∗L/(L-k+1) and G=T/N, where *N* is the mean paired-end reads coverage, *M* is the mean *k*-mer depth, *L* is the mean read length, *k* is the *k*-mer size, and *T* is the total number of base pairs. To evaluate the robustness of this method, we applied the latter formulas with different *k*-mer lengths, with k∈{17,19,21,31}. Depending on the *k*-mer length, the estimated genome size ranged from 1006.86 Mb (k=17) to 1024.35 Mb (k=31) (Appendix A). We considered the genome size estimated with k=19 (G = 1014.28 Mb) to be more reliable, as a *k*-mer of 19 is long enough to yield fairly specific genomic sequences, but also short enough to give sufficient data. Figure 1 summarizes these properties. Low coverage (<50) 19-mers with high frequency are putative erroneous *k*-mer, whereas deep coverage (>450) *k*-mer with low frequency most likely originated from repetitive genomic sequences. The 19-mer frequency graph is a bimodal distribution with two distinguishable main peaks, α (heterozygous *k*-mers) and β (homozygous *k*-mer), which suggest a low heterozygosity of the sequenced genome [21]. The heterozygosity rate, which is proportional to the ratio α/β, was roughly estimated to be 0.14% (14 SNPs per 10 kb) using the GenomeScope R-script [19]. The *k*-mers localized in single copy regions of a genome will appear uniquely in the genomic *k*-mers profile, and will fit the non-stationary portion of the *k*-mer histogram. In our case depicted in Figure 1, these are 19-mers with depth between 150 and 450. Hence, the total length of unique genomic regions (i.e., single copy portion) was estimated by the area spanned by unique *k*-mers divided by the depth of the maximal *k*-mer frequency (here β peak) [22]. Based on our 19-mer histogram in Figure 1, the single copy portion was estimated to be approximately 55% of the pikeperch genome and formalized as the following:
$$SC=\sum_{c=150}^{450}c\cdot freq_{c}/B$$
where SC is the single copy size (in bp), *c* is the *k*-mer depth, $freq_{c}$ is the corresponding frequency and *B* the depth of the main peak β. Consequently, we expect repeated sequences, including duplicated genes, interspersed and tandem repeats, to account for about 45% of the *Sander lucioperca* genome (Appendix A).

### 2.4. Genome Assembly with Long PacBio Reads

We assembled the raw PacBio single molecule sequencing reads into draft contigs using Flye vers.2.3.7 [23] with an optimized *k*-mer size of 19. Flye is a fast and accurate de novo assembler for long error-prone and noisy reads using an A-Bruijn graph to find preliminary inaccurate contigs. The inaccurate contigs are transformed into a repeats graph, which can tolerate a higher noise level than de Bruijn graphs. The long reads are then iteratively mapped back to the repeats graph to accurately resolve repeats and polish the contigs to produce the final assembly of high nucleotide-level quality. To increase the overall assembly contiguity, contigs were linked and ordered into scaffolds by mapping reads from both mate-pair libraries (2–8 kb and 2–10 kb) to contigs and by utilizing the scaffolder tool ScaffMatch v0.9 [24] to build scaffolds based on distance information from the mates. Subsequently, we used LR_Gapcloser [25] with corrected PacBio reads to fill 85% of the intra-scaffold gaps.

### 2.5. Quality Assessment of the Assembly

To evaluate the quality of the assembled pikeperch genome, we analyzed gene space completeness and reads mappability statistics, and compared this to the eight most contiguous (in terms of contigs N50 length) genome assemblies of Perciformes fishes recently published using comparable sequencing technologies and assemblies methods.

To assess the gene space completeness of this pikeperch assembly, we performed Benchmarking Universal Single-Copy Orthologs (BUSCO) analysis using BUSCO vers.3 software [26], which provides quantitative measures for the assessment of assembly completeness in regard to the expected gene content. We queried the genome against the Actinopterygii (actinopterygii_odb9, containing 4584 highly conserved single-copy core Actinopterygian genes) and Vertebrata (vertebrata_odb9, containing 2586 highly conserved single-copy core Vertebrates genes) datasets. We further evaluated the structural accuracy of the genome reconstruction by mapping genomic paired-end reads of 40 pikeperch conspecific individuals against this *Sander lucioperca* assembly, using the Burrows–Wheeler aligner (BWA), vers.0.7.17 [27].

### 2.6. Repeats Annotation

The de novo prediction of repeat elements in the *Sander lucioperca* draft genome was conducted using RepeatModeler vers.1.0.11, which comprises other tools such as RECON, RepeatScout and Tandem Repeat Finder (TRF). We also identified miniature inverted-repeat transposable elements (MITE) and long terminal repeat (LTR) retrotransposons using MITE-Tracker [28] and LTRpred pipeline respectively. Subsequently, we combined all repeats predictions—including clade specific repeats for zebrafish (*Danio rerio*), putative MITE-related sequences, full length LTR sequences and the RepeatModeler predicted library—into a comprehensive non-redundant repeats library. Finally, the combined library was mapped to the pikeperch genome using RepeatMasker vers.4.0.7 [29] to classify the transposable elements (TE).

### 2.7. Gene Structure Prediction

To annotate protein-coding genes in the pikeperch genome, we combined ab initio and homology-based methods along with RNA-Seq evidences.

For homology-based gene prediction, we obtained homologous protein sequences from seven closely related fish species, including torafugu (*Takifugu rubripes*) [30], spotted green pufferfish (*Tetraodon nigroviridis*) [31], northern snakehead (*Channa argus*) [32], red seabream (*Pagrus major*) [20], zebrafish (*Danio rerio*) [33], Antartic dragonfish (*Parachaenichthys charcoti*) [34], and Chinese sillago (*Sillago sinica*) [35]. We used TBLASTN vers.2.5.0 [36], with an e-value cutoff of 1e-6 to align these homologous protein sequences to the *Sander lucioperca* genome. For each protein sequence, we retained only the top scoring alignments with a minimum identity of 80%. Exonerate vers.2.4.0 [37] was then used to map these top scoring proteins to the *Sander lucioperca* genome in order to predict putative gene models.

In the *ab initio* approach, we applied AUGUSTUS vers.3.2.3 [38], and GENSCAN vers.1.0 [39] to predict gene structures on the repeat-masked genome. While Augustus was trained with randomly selected full-length protein-coding genes as predicted by Exonerate, GENSCAN was run with human parameters.

The transcript-based gene prediction was performed using RNA-Seq data of a conspecific individual, whose paired-end reads were obtained from the Sequence Read Archive (SRA), (Accession-Nr: SRR2871497). These reads were mapped to our pikeperch genome using HISAT2 vers.2.1 [40], a splice-aware aligner, to detect splice junctions. Cufflinks vers.2.2.1 [41] was subsequently used to assemble transcripts based on HISAT2 alignments. In addition, we generated a de novo assembly from the same RNA-Seq data using Trinity vers.2.8.4 [42]. Finally, we retained only transcript sequences that were predicted by both approaches and that had at least 99% identity over their full-length.

The gene models prediction from the three methods were integrated using EvidenceModeler [43], to build a consensus, non-redundant *Sander lucioperca* gene set. Ultimately, the resulting gene set was filtered to remove genes that had no start and/or stopcodon, or had an in-frame stopcodon, or had a coding sequence (CDS) shorter than 150 nt.

Finally, we annotated three types of non-coding RNAs (ncRNAs) in the pikeperch genome with methods specific to each type of ncRNAs. Transfer RNA (tRNAs) were predicted using tRNAscan-SE vers.2.0 with eukaryote parameters [44]. Eukaryotic ribosomal RNAs (rRNAs) were annotated utilizing the software package RNAmmer vers.1.2 [45], and putative micro RNAs (miRNAs) were predicted by homology to the known mature miRNAs sequences available in the miRBASE database [46], by using the miRDeep2 pipeline [47].

### 2.8. Preliminary Functional Annotation of Protein-Coding Genes

For preliminary functional annotation of predicted genes, the pikeperch protein-coding sequences were mapped against different functional databases including SwissProt, TrEMBL, and the NCBI non-redundant (NR) protein database using BLAST with an e-value cutoff of 1e-5. To identify known protein domains and motifs, the CDS were also searched against all entries in the InterPro dataset v.73 [48] using InterProScan vers.5 [49].

### 2.9. Gene Orthologs Analysis

In order to identify gene families among selected Perciformes fish species, orthogroups were identified using OrthoFinder vers.2 [50]. The coding sequences of Chinese sillago, northern snakehead, Antartic dragonfish, and spotted sea bass (*Lateolabrax maculatus*) were collected from GigaDB [51] in their respective repository. The Coding sequences of yellow perch (*Perca flavescens*) and red seabream were obtained from NCBI’s ENTREZ database with, respectively, *SAMD00076252* and *SAMN10722690* as BioSample-ID. To infer orthologous gene families, the 185,203 CDS (proteins) of all seven species were aligned in an all-vs.-all fashion using BLASTP with an e-value threshold of 1e-5. The BLASTP alignments were fed to the OrthoFinder algorithm, which applied the Markov Cluster Algorithm (MCL) to cluster alignments into 18,917 orthogroups (families). We have subsequently constructed the phylogenetic tree of all seven species based on the 1:1 single copy orthologous genes clusters. For each single-copy cluster (i.e., family), and for each species, single-copy orthologous genes were concatenated into a super-gene, and multiple sequence alignments (MSA) were generated using mafft vers.7 [52]. The rooted species tree was inferred from the generated MSA using approximately-maximum-likelihood methods implemented in FastTree [53]. Finally, the MCMCtree program in the PAML package [54] was used to estimate the divergence time in each tree node with the approximate likelihood method and the Jukes–Cantor substitution model. The molecular clock data from the divergence time between red seabream and Chinese sillago provided in the TimeTree database [55] were used for root calibration.

## 3. Results

We employed a whole genome shotgun (WGS) strategy to produce 412.8 Gb (350X genome coverage), 74.2 Gb (66X genome coverage) and 71.4 Gb (63X genome coverage) of data corresponding to data yielded by Illumina paired-end, 2–8 kb and 2–10 kb mate-pairs libraries, respectively. In addition, 66 Gb (60X genome coverage) data were generated with size selected 20 kb PacBio libraries. The mean reads length for Illumina data was 150 bp. The PacBio data had a mean read and N50 length of 12.7 kb and 16.4 kb, respectively (Appendix A). The paired-end reads were primarily used for genome properties estimation, to assess and improve the base-level quality of the assembly. Our estimations based on *k*-mer analysis have shown that the pikeperch genome is as large as 1014 Mb, which is consistent with the previous estimate of 1114 Mb, based on cytometric methods [6]. The *k*-mer analysis also revealed that, we could expect about 45% of repetitive DNA sequences, since the single copy portion in the pikeperch genome was roughly estimated to be 55%. The Illumina long-insert reads (2–8 kb and 2–10 kb) were used for scaffolding, while the PacBio data were exclusively utilized to produce the contig-scale assembly and fill over 90% of the intra-scaffold gaps.

We assembled the PacBio reads into a final assembly size of ~900 Mb covering 89% of the 1,014 Mb estimated genome size. The draft genome preliminary consisted of 1,966 contigs with a N50 length of 3.0 Mb (Figure 2). In particular, 75.8% of the genome is covered by 207 contigs larger than 1 Mb, and only 3.9% of the genome is spanned by contigs shorter than 100 kb. The contigs were ordered into 1,313 scaffolds with N50 size of 4.9 Mb, representing an increase of 63% in contiguity over to the contig-level assembly (Table 2). The largest contig and scaffold was 17.7 Mb and 19.0 Mb long, respectively, which might span a full chromosome arm. Hence, this assembly is more contiguous than most of the newly published Perciformes fish genomes as depicted in Figure 3.

In total, the repetitive elements accounted for 352 Mb, representing 39% of the *Sander lucioperca* genome. DNA transposons (136 Mb) were the most predominant type of repeats, accounting for 15.2% of the the assembled genome and 72.8% of all identified transposable elements (TEs) (Appendix A). We correlated the repeat content with the genome size of the most contiguous assemblies of Perciformes species, which have recently been published, assuming that a high positive correlation might support a coherent prediction of *Sander lucioperca* repeat content. As expected, we found a strong correlation ($Pearson^{\prime }s\;R=0.91,\;p=0.00065$) between repeat content and genome size (Figure 2). In particular, *Sander lucioperca* has the largest genome size and repeat content among the compared Perciformes.

The evaluation of the structural accuracy highlighted that, more than 99.9% of the Illumina paired-end genomic reads of a pikeperch population (40 individuals) were reliably aligned to our PacBio-based assembly. Moreover, approximately 97% of these reads have been properly aligned with the correct distance to their mates (Figure 3C). This high mapping rate and alignment accuracy of the read pairs not only demonstrate the high structural accuracy of the contigs, but also indicate that the assembly is nearly complete in terms of genome coverage. This claim is substantiated by the gene space completeness and connectivity assessment of the assembly using BUSCO. By querying against both the Actinopterygii (4584 core genes) and Vertebrata (2586 core genes), we found that, 96.3% and 97.6% of core genes, respectively, were identified in full-length as single-copy in this pikeperch assembly (Table 1). Additionally, 89 (1.945%) Actinopterygians and 40 (1.54%) Vertebrates core genes were captured, though fragmented. This suggests, that less than 1.6% of the core Vertebrates and Ray-finned fish genes were missing in our pikeperch assembly.

The gene model prediction resulted in 21,249 protein-coding genes with an average CDS of 1,313 bp and 6.7 exons per CDS. (Table 2). These genes are scattered over 828 scaffolds, averaging 25.6 genes per scaffold. Most of them (87%) had significant matches with at least one InterPro database. Moreover, 64.8% of the predicted genes were associated with at least one functional entry in te Swissprot database; 87.2% had significant TrEMBL database hits; and 87.2% were significantly mapped to NCBI RefSeq non-redundant proteins (NR) (Table 2). The more noteworthy was that, around 60% (10,980) of NR top hits were homologous to RefSeq genes annotated in yellow perch using the NCBI Eukaryotic Genome Annotation Pipeline. In addition, a total of 2,659 putative ncRNAs were predicted—including 2,313 tRNAs, 180 rRNAs and 166 miRNAs (Table 2).

To relatively integrate *Sander lucioperca* in the Perciformes clade, protein sequences of pikeperch along with six closely related Perciformes fishes were used to predict orthogroups. The closely related species consisted of Chinese sillago, northern snakehead, Antartic dragonfish, spotted seabass, yellow perch and red seabream species. A total of 18,917 orthogroups (gene families) were predicted, of which 1,221 (6.4%) were 1:1 single-copy. Moreover, 239 gene families were pikeperch-specific (Figure 4A). Among the compared species, Yellow perch had the largest number of shared gene families (16,188), while pikeperch had the smallest number (9,078) of shared gene families (Figure 4B). Phylogenetic analysis using 1:1 single-copy orthologs between these species, suggested that the closely related pikeperch and yellow perch, which belong both to the Percidae family, diverged from their last common ancestor around 35 million years ago (Figure 5). As expected, the two Percid species shared the maximum number (454) of orthogroups, when comparing all species pairwise.

## 4. Discussion and Conclusions

*Sander lucioperca* is one of the fresh and brackish water fish species that has recently shown a particular promise in the aquaculture industry in Europe. This emerging aquaculture species is particularly valued for its good growth performance, its highly priced meat which contains only few intermuscular bones, and its high protein content. However, the lack of omics data, in particular a genome sequence, has been hindering the understanding of genetic factors associated with growth, performance and adaptability of this fish in captive conditions. In this study, we have successfully sequenced, assembled and annotated the first draft genome of the pikeperch using PacBio long reads from the third-generation sequencing, and taking advantage of the Illumina short reads accuracy.

The quality and accuracy of a genome assembly is assessed by state-of-the-art approaches such as the completeness of lineage-specific single-copy orthologs, estimating the mapping rate of genomic reads, or comparing the assembly and annotation metrics with those of closely related species [56,57]. Our reported assembly has only 1966 contigs and 1313 scaffolds with 3 Mb and 4.9 Mb as contigs and scaffolds N50, respectively. These are outstanding contiguity metrics compared to recently reported assemblies of other Perciformes (Figure 2, Appendix A), which have even smaller genomes with fewer repeats, thus a less challenging assembly—at least theoretically. Interestingly, 99.9% of DNA PE-reads from a population of 40 conspecific pikeperch individuals were aligned to this draft genome, of which 97% of the read pairs were mapped concordantly. That is, the forward and reverse reads were consistently aligned, respecting their inner distance and relative orientation as defined by the insert library. Since the assembly was generated independently of these PE-reads, the outstanding rate of concordantly mapped paired-read is indicative of a highly contiguous and structurally accurate assembly. This is substantiated by BUSCO metrics on Vertebrata and fish-specific Actinopterygii datasets. In particular, approximately 97.56% of Vertebrates and 96.26% of Actinopterygians core genes were captured as complete single-copy orthologs in our assembly. This score even exceeds 98.5% if we consider fragmented core genes, which were also captured. Compared to assemblies of closely related Percids, which have comparable genome size, our pikeperch assembly has 50 times fewer contigs than the Eurasian perch (*Perca fluviatilis*), and only two times more contigs than the yellow perch (Table 3) [58].

Overall, this is evidence that our reported genome is structurally accurate. Particularly, the gene-rich regions have been accurately sequenced and assembled. The proportion and content of protein-coding genes in pikeperch (21,249) is comparable with those predicted in other recently published Perciformes genomes, including Chinese sillago (22,122) [35], Eurasian perch (23,397) [58], yellow perch (23,749) [59], northern snakehead (19,877) [32], and spotted sea bass (22,015) [60]. The phylogenetic analysis based on 1:1 single copy orthologs among selected Perciformes species showed that the yellow perch and pikeperch share the maximum number of gene families. This is due to the fact that they are genetically and taxonomically closer than the other Perciformes species. This phylogenetic classification is also consistent with the prediction reported in previous studies [61,62].

In summary, the draft assembly and the sequencing data we report here are the most awaited genomic resources to pave the way for genomic studies such as genotyping by sequencing, genetic selection and diversity on pikeperch. Such studies will provide an impetus for the industrial production of this species. The gene annotations we report in this study provide the first overview of the gene content in pikeperch. It will enhance subsequent functional genomic analyses of molecular markers associated with key phenotypic features and is relevant for marker-assisted breeding.

## Figures and Tables

**Figure 1 genes-10-00708-f001:**
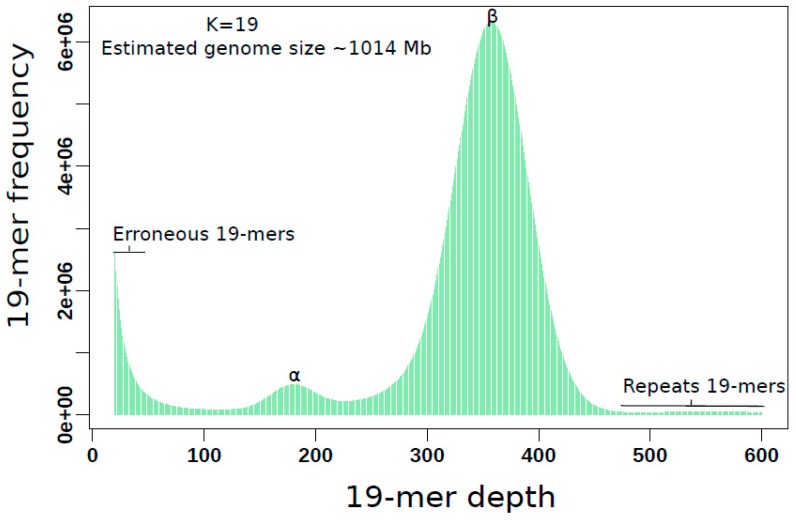
Estimated characteristics of *Sander lucioperca* genome based on 19-mer analysis. The vertical axis represents the 19-mer depth, and the horizontal their corresponding frequency. α is the heterozygous and β the homozygous peak. Low coverage (<50) 19-mers are putative erroneous sequences, whereas deep coverage (>450) 19-mer indicate repetitive genomic sequences.

**Figure 2 genes-10-00708-f002:**
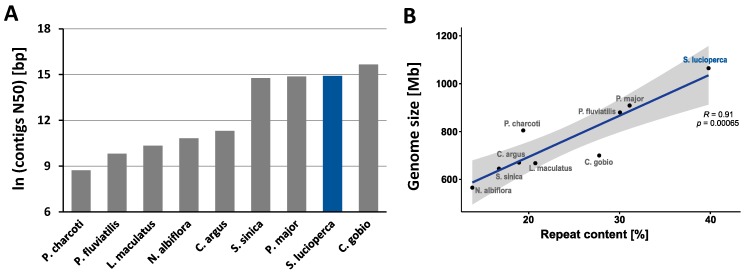
Comparison of contiguity (N50) and repeat content among selected Perciformes fish species. (**A**): Contigs N50 (scaled with natural logarithm) of the pikeperch assembly compared with recently published assemblies of species of the same taxonomic order (Perciformes). (**B**): Correlation of repeat content and genome size in recently published genomes of Perciformes fish species. *R* is the Pearson’s correlation coefficient and *p* the associated *p*-value.

**Figure 3 genes-10-00708-f003:**
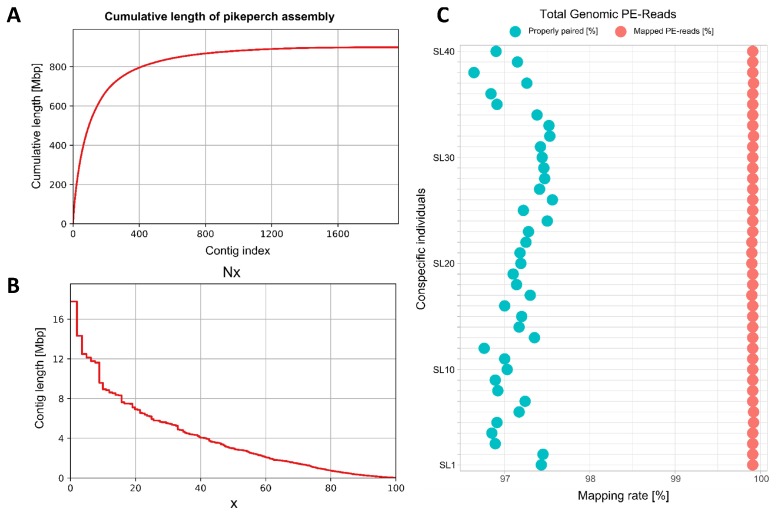
Assembly length and mappability statistics. (**A**): The cumulative length of pikeperch assembly in correlation with the total number of contigs, sorted from the largest to the shortest. (**B**): Overall trend of contigs Nx-metric as x varies from 0 to 100. (**C**): Mapping rates of genomic paired-end reads of 40 pikeperch individuals to our constructed reference pikeperch assembly.

**Figure 4 genes-10-00708-f004:**
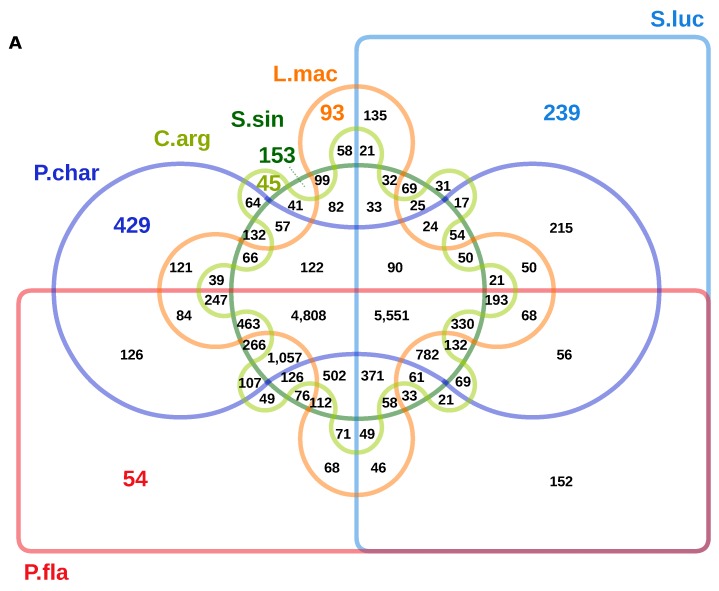
Shared gene families and their distribution per species. (**A**): Venn-diagram showing the shared gene families between selected Perciformes species: L.mac (*Lateolabrax maculatus*), S.sin (*Sillago sinica*), C.arg (*Channa argus*), P.fla (*Perca flavescens*), S.luc (*Sander lucioperca*), P.char (*Parachaenichthys charcoti*). Colored numbers indicate the number of species-specific gene families. (**B**): Total number of gene families for each species.

**Figure 5 genes-10-00708-f005:**
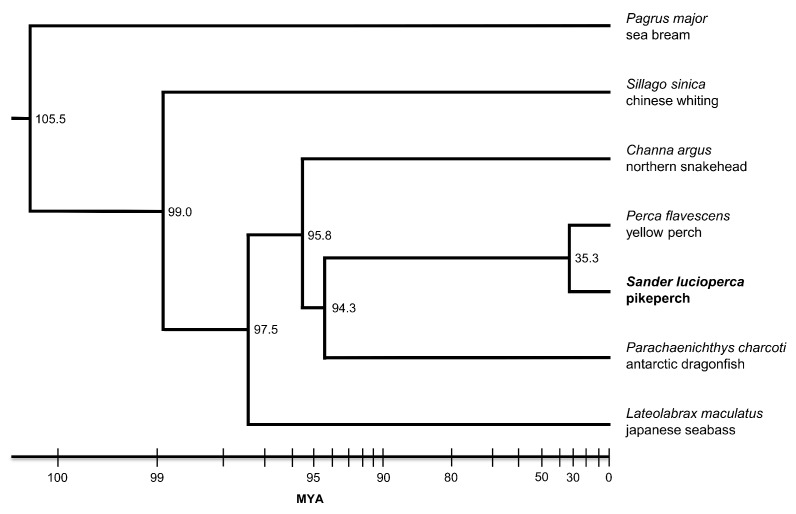
Phylogenetic analysis of *Sander lucioperca* and closely related Perciformes genomes. The constructed phylogenetic tree is based on one-to-one single-copy orthologs between the seven Perciformes fish species. The node labels indicate the estimated divergence time from the last common ancestor (LCA), in million years ago (MYA).

**Table 1 genes-10-00708-t001:** Summary statistics of Benchmarking Universal Single-Copy Orthologs (BUSCO) analysis for *Sander lucioperca* genome assembly.

Categories	Actinopterygii		Vertebrata	
	#Genes	Percentage	#Genes	Percentage
Complete single-copy	4413	96.27	2523	97.56
Complete duplicated	112	2.45	26	1.01
Fragmented	89	1.94	40	1.54
Missing	82	1.79	23	0.89

**Table 2 genes-10-00708-t002:** Summary statistics of *Sander lucioperca* genome assembly and annotation.

**A-ASSEMBLY**	
Total size (nt)	900,477,756
No. of contigs	1966
Contigs N50 (nt)	2,995,800
Longest contig (nt)	17,774,792
No. of scaffolds	1313
Scaffold N50 (nt)	4,929,547
Longest scaffold (nt)	19,065,786
Average scaffold (nt)	685,817
GC-content (%)	40.91
**B-PROTEIN-CODING GENES**	
Number of coding genes	21,249
mean gene length (nt)	10,961
Mean coding sequence (CDS) length (nt)	1313
Mean intron length (nt)	1696
Mean exon length (nt)	196
Average no. of exons per CDS	6.7
% of genome covered by genes	25.9
% of genome covered by CDS	3.1
**C-FUNCTIONAL DATABASES**	
Non-redundant (NR) hits	18,536 (87.2%)
Swissprot hits	13,783 (64.8%)
trEMBL hits	18,171 (85.5%)
Interpro hits	18,486 (87.0 %)
**D-NON-CODING RNA PREDICTION**	
tRNA	2313
rRNA	180
miRNA	166

**Table 3 genes-10-00708-t003:** Comparison of currently reported genome assemblies of fish species in the Percidae family.

	Estimated Repeat Content (%)	Total Assembly Length (Mb)	Ungapped Length (Mb)/(%)	Number of Contigs	Contigs N50 (Mb)	#Coding Genes
Yellow perch	41	877.4	877.0 (99.9%)	1097	4.2	23,749
Pikeperch	39	900.5	899.8 (99.9%)	1966	3.0	21,249
Eurasian perch	33	958.2	851.6 (88.9%)	100,821	0.0182	23,397

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
