# Peer review of "The First Highly Contiguous Genome Assembly of Pikeperch (Sander lucioperca), an Emerging Aquaculture Species in Europe"

_genes, 2019, doi:10.3390/genes10090708_

Round 1

Reviewer 1 Report

This is an interesting and very well written paper. I was not able to find indications about how the single copy size of the genome was computed but otherwise I cannot find any criticism.

Reviewer 2 Report

The manuscript “The first highly contiguous genome assembly of pikeperch (Sander lucioperca), an emerging aquaculture species in Europe” by Nguinkal and others describes the sequencing and genome assembly of the freshwater fish pikeperch. More and more genomes are becoming available and this will help both evolutionary biologists as well as biotechnological goals. This bioinformatics manuscript describes the basic features/statistics of the modestly sized genome of pikeperch in relationship other assembled fish genomes. It will be of interest for their future goals to learn if pikeperch can be easily manipulated with technologies such as CRISPR. Overall, it appears a solid assembly. There are only a few concerns that need to be addressed.

Major concern:

+ The accession numbers for the raw sequence data as well as contigs must be provided.

Minor concerns:

+ Does S lucioperca have XY or ZW sex chromosomes? If it is the latter, sequencing and assembling a female genome should be considered.

+ Figure 1: please label the axes in the figure.

+ there are some double spacings between sentences here and there.

+ the authors could comment on the presence or absence of unique genes or gene sets, or recently duplicated genes. This information could help with the future goals laid out by the authors.

+ typo line 66: “20 Kb” should be 20 kb.

+ typo lines 179-180: “1:1 s A high-quality” and “ingle copy” should be Single copy.

+ typo line 238: “round” should be around or approximately.
